# Model Predictive Secondary Frequency Control for Islanded Microgrid under Wind and Solar Stochastics

Zhongwei Zhao [1,*], Xiangyu Zhang [2] and Cheng Zhong [2]

[1] College of Information and Electronic Engineering, Zhejiang Gongshang University, Qiantang District, Hangzhou 310018, China

[2] College of Electrical Engineering, Northeast Electric Power University, Jilin 132012, China; m15264388175@163.com (X.Z.); zhongcheng@neepu.edu.cn (C.Z.)

* Correspondence: zhaozw@zjgsu.edu.cn

**Abstract:** As microgrids are the main carriers of renewable energy sources (RESs), research on them has been receiving more attention. When considering the increase in the penetration of renewable energy sources/distributed generators (DGs) in microgrids, their low inertia and high stochastic power disturbance pose more challenges for frequency control. To address these challenges, this paper proposes a model predictive control (MPC) secondary control that incorporates an unknown input observer and where RESs/DGs use a deloading virtual synchronous generator (VSG) control to improve the system's inertia. An unknown input observer is employed to estimate the system states and random power disturbance from the RESs/DGs and load to improve the effect of the predictive control. The distributed restorative power of each DG is obtained by solving the quadratic programming (QP) optimal problem with variable constraints. The RESs/DGs are given priority to participate in secondary frequency control due to the proper weighting factors being set. An islanded microgrid model consisting of multiple photovoltaic and wind power sources was built. The simulation results demonstrate that the proposed method improves the system frequency, restoration speed, and reduces frequency deviations compared with the traditional secondary control method.

**Keywords:** secondary frequency control; model predictive control; virtual synchronous generator; unknown input observer; islanded microgrid

## 1. Introduction

Microgrids have attracted tremendous attention because they can facilitate the integration of RESs/DGs and improve the reliability, efficiency, and flexibility of power grids [1]. Microgrids can be operated in an islanded mode, wherein the primary objective is to maintain a balance between sources and loads. RESs/DGs are connected to the microgrid through inverters, which makes the microgrid less inertial than the traditional grid and deteriorates the frequency characteristics of the system [2]; additionally, due to the uncertainty of RESs, more unknown disturbances are brought into the system [3]. All of these factors bring more challenges to the frequency regulation of microgrids [4].

Hierarchical control is the basic control strategy for microgrids, which includes primary frequency regulation, secondary frequency regulation, and even tertiary frequency regulation [5,6]. Primary regulation immediately adjusts the power output from the local governor or electronic controller to address the microgrid's frequency deviation. The conventional primary control is a droop control, but it provides barely any inertia/damping support. To cope with this problem, the concept of the VSG has been presented. VSG mimics the behavior of conventional synchronous generators (SGs) and achieves primary frequency control while compensating for some inertia deficiency [7]. The primary control cannot achieve error-free regulation [8] and must introduce the secondary control; this is a control that utilizes measurements and communication systems to eliminate frequency deviation [9].

Many control methods have been utilized for the secondary frequency control of islanded microgrids [10]. The authors in [11] adopted a sliding mode control to control frequency; this method can overcome system uncertainty and has strong robustness, but the chattering phenomenon exists in sliding mode controls. The linear quadratic regulator (LQR) has also been used for the frequency control of microgrids [12,13]. Its design is convenient and suitable for MIMO systems, but it requires a high precision of the control parameters. Some artificial intelligence (AI) methods are also used for frequency control. In [14,15], the black hole optimization search algorithm and particle swarm optimization algorithm were used for secondary frequency control. The authors in [16,17] used a fuzzy adaptive controller and self-tuning techniques based on AI for secondary frequency control. However, these AI-based methods do not consider the control quantity in the objective function, which may cause power overlimit and waste renewable energy. In addition to centralized schemes, the distributed control has been previously adopted [18–20], but distributed control has the problems of high technical requirements, high costs, traffic congestion, and limits on critical monitoring and protection functions [21].

The emerging MPC method has a good control effect and robustness, and the rolling optimization algorithm can make up for the uncertainty that is caused by model mismatch, distortion, interference, and other factors [22]. The study in [23] is a review of the application of MPC in microgrids. The study analyzed the application of MPC in microgrids, covering various levels of the hierarchical control structure; however, it did not focus on the application of MPC in microgrid frequency regulation. The authors in [24] presented a novel secondary control method for voltage regulation in islanded microgrids. In [25], an MPC method for a renewable energy AC microgrid without a PID regulator was proposed. A tube-based MPC method was presented in [26] to control the participation of electric vehicles (EVs) in the frequency regulation of an islanded power system. The authors in [27] presented a modified MPC strategy to account for limiting overcurrent in the case of a faulted autonomous AC microgrid operation; however, this fails to address the impact of RES randomness. In [28], the authors considered energy storage systems based on virtual inertia control in coordination with a diesel generator that was controlled by the MPC method to improve the frequency regulation of an islanded MG. The authors in [29] proposed a fuzzy control combined with an MPC for VSG control, where the fuzzy MPC controller enhances the frequency stability of islanded microgrids by adjusting virtual inertia and damping coefficients. The authors in [30] presented an MPC strategy that optimizes the power flow between energy storage batteries in a microgrid and employs a solver to handle the nonlinear changes in energy storage batteries. However, the above-mentioned MPC studies only utilize storage batteries and diesel generators for participating in frequency regulation. Recently, RESs have been proposed to participate in system frequency regulation [31,32] through a deloading control that the DGs operate at suboptimal points. MPC requires an observer to estimate the current system state. The uncertainties from the DGs may deteriorate the performance of the observer. Conventional observers struggle to obtain accurate values because they only use input signals, including perturbed signals [33].

This paper proposes a load frequency control strategy for islanded microgrids based on the MPC method. The main salient features of the proposed method are briefly outlined as follows: (1) RESs adopt a deloading control and VSG control to provide reserve power, virtual inertia, and participation in secondary frequency regulation; (2) The decoupled unknown state observer is used to observe the system state value and power disturbance, respectively; (3) MPC is utilized to optimize the output of each RES/DG for secondary frequency control in real time. RESs/DGs are given priority by setting the appropriate weight factors, and the variable constraints are set based on the maximum available power of RESs/DGs to prevent their output power from exceeding the limit.

The rest of this article is organized as follows. Section 2.1 includes the study of microgrid load frequency modeling and its space state equations. Section 2.2 provides the unknown input observer design. The proposed MPC controller is described in Section 2.3. Section 2.4 focuses on wind power, photovoltaic, and load uncertainty modeling. The

simulation and results are demonstrated in Section 3, and the conclusion is presented in Section 4.

## 2. Materials and Methods

### 2.1. Microgrid Model

The islanded microgrid studied in this paper is shown in Figure 1. It mainly contains two distributed photovoltaic generators (PV1, PV2), two distributed wind turbines (WT1, WT2), a battery energy storage system unit (BESS), and a diesel engine unit (DU). The capacity configuration of each power generation unit is shown in Table 1 below.

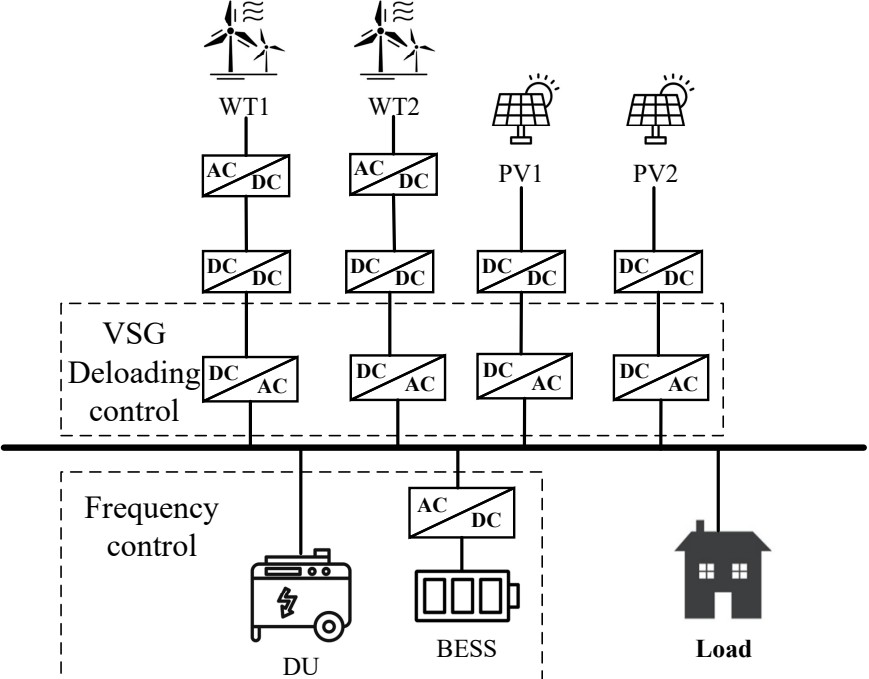

**Figure 1.** Schematic diagram of the microgrid structure.

**Table 1.** Microgrid-rated power and load.

| Microgrid Power | Symbols | Numerical Value |
|---|---|---|
| Photovoltaic (PV) | $P_{PV1}$ | 100 kW |
| | $P_{PV2}$ | 100 kW |
| Wind turbine (WT) | $P_{wt1}$ | 100 kW |
| | $P_{wt2}$ | 100 kW |
| BESS | $P_{BESS}$ | 120 kW |
| Diesel engine | $P_{DU}$ | 120 kW |
| Load | Load | 640 kW |

The conventional primary frequency control is droop control, but it provides barely any inertia/damping support. In Figure 1, the VSG deloading control is used for grid-tied converters to achieve virtual inertia and primary frequency regulation and obtain partial reserve power. It should be noted that RESs can participate in secondary frequency regulation by using a deloading control, but this also generates the influence of input uncertainty and affects the control effect. Furthermore, the stochastic power fluctuation of the WT, PV, and load causes system frequency fluctuation.

#### 2.1.1. The Small Signal of VSG

In this study, the RESs adopt a VSG deloading control. The DC side provides a stable voltage for the inverter. The power command value is set according to the maximum avail-

able power and deloading coefficient. Deloading control has been extensively researched and is not the focus of this paper; however, [34] can serve as a reference for it.

In Figure 2, $I_{dc}$ [A] and $U_{dc}$ [V] are the DC side current and voltage, respectively. $E$ [V] is the inverter output voltage, $U_l$ [V] is the grid voltage, $I$ [A] is the grid current, $L_f$ [mH] is the filter inductor, $C_f$ [mF] is the filter capacitor, $L_g$ [mH] is the line inductor, $K_q$ is the voltage sag factor, $P_{in}$ [kW] is the input power of the inverter, $P_{in}$ [kW] is the output power of the inverter, and $P_{ref}$ [kW] and $Q_{ref}$ [kVar] are the command values of active and reactive power, respectively.

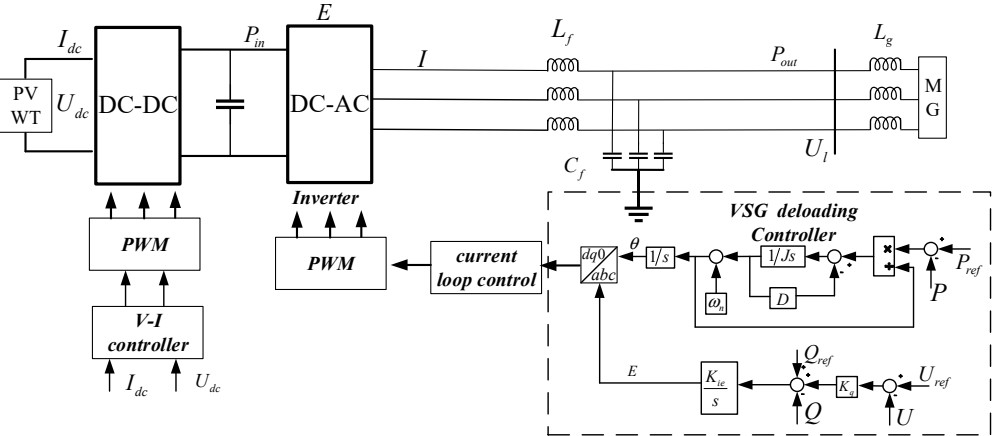

**Figure 2.** VSG control schematic diagram.

The mechanical equation of the synchronous generator is

$$J\omega\frac{\mathrm{d}\omega}{\mathrm{d}t} = P_\mathrm{s} - P_\mathrm{e} - D\omega(\omega - \omega_n) \tag{1}$$

where $P_s$ [kW] is the mechanical power, $P_e$ [kW] is the electromagnetic power, and $D$ is the damping factor of the generator. $J$ is the rotational inertia of the generator, and $\omega$ [rad/s] is the mechanical angular velocity of the rotor; the electrical angular velocity is also $\omega$ [rad/s], and $\omega_n$ [rad/s] is the synchronous angular velocity of the grid.

The active output of the inverter can be expressed as

$$P = \frac{EU_l}{Z}\cos(\varphi - \delta) - \frac{U_l^2}{Z}\cos\varphi \tag{2}$$

where $Z$ [$\Omega$] is the line impedance, $P$ [kW] is the active output power of the inverter, $E$[V] is the inverter output voltage, $U_l$ [V] is the grid voltage, $\varphi$ [rad] is the line impedance angle, and $\delta$ [rad] is the power angle.

For simplicity, the line impedance is usually represented as purely inductive and $\delta$ is a small value; thus, Equation (2) can be simplified as

$$P = 3\frac{UU_l}{\omega L}\delta \tag{3}$$

where $U$ [V] is the inverter-side voltage, $L$ [mF] is line inductive reactance, and $U_l$ [V] is the grid-side voltage.

From Equation (3), the output power and power angle are linearly related. The power angle of VSG can be obtained through the rotor mechanical Equation (1).

By combining Equations (1) and (3), the VSG small-signal model can be obtained as shown in Figure 3, where $K = 3\frac{UU_l}{\omega L}$.

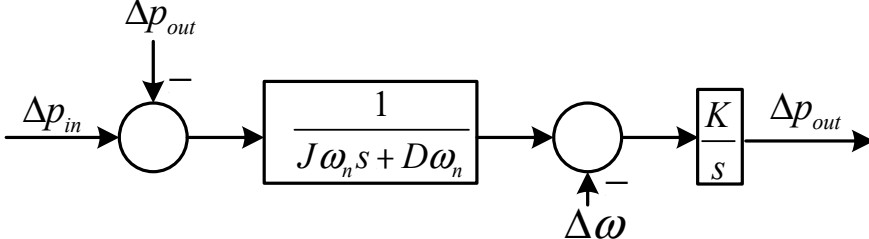

**Figure 3.** Virtual synchronous generator small-signal model.

### 2.1.2. LFC Model of Microgrid

This paper mainly focuses on the secondary frequency control of RES high-penetration microgrids. The load frequency control (LFC) model of the microgrid is shown below in Figure 4. In the LFC model, the VSG small-signal model is used to simulate the grid-connected inverter. The energy storage battery and the diesel engine adopt the first-order equivalent model. More details can be found in [35]. In order to facilitate the research, all parameters are normalized.

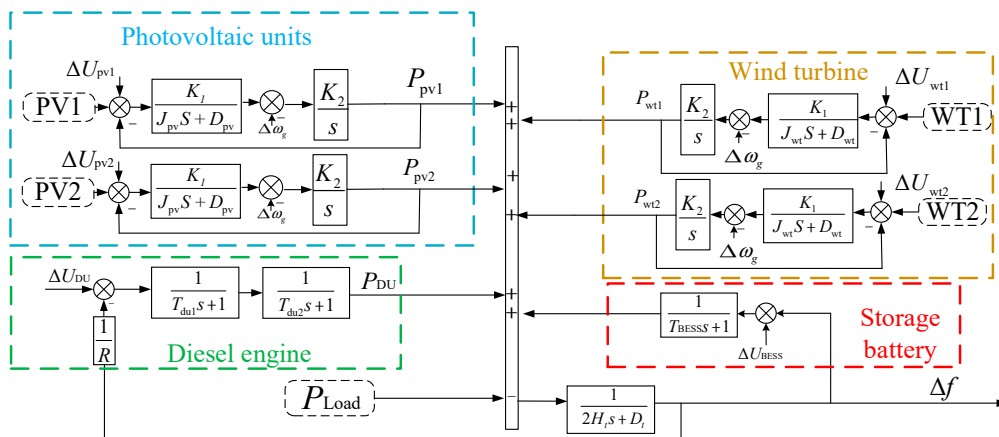

**Figure 4.** Equivalent model of the microgrid.

In Figure 4, $J_{pv}$ and $J_{wt}$ are the inertia values of the virtual synchronous generator of the photovoltaic and wind turbines, respectively. $D_{pv}$ and $D_{wt}$ are the damping values of the virtual synchronous generator of the photovoltaic and wind turbines, respectively. $\Delta U_{pv}$, $\Delta U_{wt}$, $\Delta U_{du}$, and $\Delta U_{bess}$ are the power reference values of the photovoltaic, wind turbine, diesel engine, and energy storage battery, respectively. $R$ and $T_{du1}$ are the diesel sag factor and speed regulation time constant, respectively, and $T_{du2}$ is the time constant of the diesel engine. $K_1$ is the rated angular velocity, $K_2 = 3\frac{UU_g}{\omega L}$, $H_t$ is the microgrid inertia, and $D_t$ is the damping coefficient of the system. The specific values of the LFC microgrid modeling are listed in Table 2.

**Table 2.** Microgrid parameters.

| Microgrid Power | Symbols | Value |
|---|---|---|
| Photovoltaic (PV) | $J_{PV}$ | 0.0987 |
| | $D_{PV}$ | 88.8264 |
| Wind turbine | $J_{wt}$ | 0.0987 |
| | $D_{wt}$ | 88.8264 |
| Energy Storage System | $T_{BESS}$ | 0.01 |
| Diesel engine | $T_{du1}$ | 0.04 |
| | $T_{du2}$ | 0.01 |
| | $R$ | 3 |
| Microgrid inertia | $H_t$ | 0.6 |
| Microgrid damping | $D$ | 1 |

2.1.3. State-Space Equation Modeling

The microgrid LFC model of Figure 4 is organized into a state-space form as follows:

$$\begin{cases} \dot{x}(t) = Ax(t) + Bu(t) \\ y(t) = Cx(t) \end{cases} \tag{4}$$

where $x(t)$ is the state variable, $u(t)$ is the control variable and unknown input perturbation, and $y(t)$ is the output variable; $A$, $B$, and $C$ are the state constant matrix, control constant matrix, and output constant matrix of the continuous state equation, respectively. The system state and control quantities are shown below.

$$x = \begin{pmatrix} \Delta\omega_{pv1} & \Delta p_{pv1} & \Delta\omega_{pv2} & \Delta p_{pv2} & \Delta\omega_{wt1} & \Delta p_{wt1} & \Delta\omega_{wt2} & \Delta p_{wt2} & \Delta p_{bess} & \Delta\omega_{du} & \Delta p_{du} & \Delta f \end{pmatrix}^T$$

$$u = \begin{pmatrix} \Delta u_{pv1} & \Delta u_{pv2} & \Delta u_{wt1} & \Delta u_{wt2} & \Delta u_{bess} & \Delta u_{du} \end{pmatrix}^T$$

where $\Delta\omega_{pv1}\Delta\omega_{pv2}$ is the angular velocity of the PV virtual synchronous generator, $\Delta p_{pv1}$ $\Delta p_{pv2}$ is the PV output power, $\Delta\omega_{wt1}$ $\Delta\omega_{wt2}$ is the angular velocity of the WT virtual synchronous generator, $\Delta p_{wt1}$ $\Delta p_{wt2}$ is the WT output power, $\Delta\omega_{bess}$ is the angular velocity of the BESS virtual synchronous generator, $\Delta p_{bess}$ is the BESS output power, $\Delta p_{du}$ is the diesel engine output power, $\Delta\omega_{du}$ is the diesel engine angular velocity, and $\Delta f$ is the system frequency variation. $\Delta u$ is the control variable of DGs.

*2.2. Unknown Input Observer Design*

In this study, an MPC is used for secondary frequency control because it can achieve satisfactory control effects when dealing with multi-input multi-output coupled systems. MPC uses the current system state as the initial value for the predictive control. However, the system's current state is not directly measurable, so a state observer is necessary.

Due to the randomness of the system, conventional observers struggle to obtain accurate system states. This paper uses an unknown input observer to obtain the state estimation and unknown input perturbation (the equivalent load disturbance).

The equivalent load disturbance is represented as an unknown input disturbance $d_k$. The state equations of the microgrid can be rewritten as

$$\begin{cases} x_k = A_{ob} * x_{k-1} + B_{ob} * u_k + E_{ob}d_k \\ y_k = C_{ob}x_k \end{cases} \tag{5}$$

where $A_{ob}$ is the state matrix of the system; $B_{ob}$ is the control matrix of the system, which does not contain unknown perturbation terms; $C_{ob}$ is the output matrix of the system; $E_{ob}$ is the matrix of unknown input perturbations; $x_k$ is the state vector; $y_k$ is the output vector; $u_k$ is the known input vector; and $d_k$ is the equivalent load perturbation. They are all discretized matrices.

The state estimation error vector $e_k$ is defined as

$$e_k = x_k - \hat{x}_{(k|k)} \tag{6}$$

where $\hat{x}(k|k)$ is the state estimate.

The structure of the UIO according to the dynamic system design is shown in Figure 5. The equations are as follows:

$$z_k = F * z_{k-1} + TBu_{k-1} + K * y_k \tag{7}$$

$$\hat{x}_{(k|k)} = z_k + H * y_k \tag{8}$$

where $z_k$ is the state of the observer, and $F$, $T$, $K$, and $H$ are the matrices designed to achieve the decoupling of the unknown inputs. From the block diagram, it can be seen that the UIO

is essentially a dynamic system that decouples the state estimates from the perturbation terms in the original system.

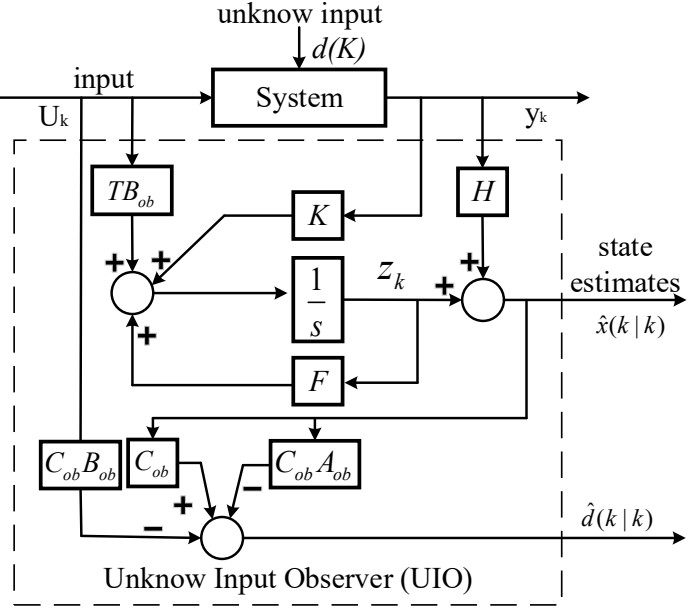

**Figure 5.** UIO schematic diagram.

Expanding $\dot{e}(k)$, we obtain:

$$
\begin{aligned}
\dot{e}(k) = {}&(A - HCA - K_1C)e(k) + [F - (A - HCA - K_1C)]z(k) \\
&+ [K_2 - H(A - HCA - K_1C)]y(k) + [T - (I - HC)]Bu(k) + (HC - I)Ed(k)
\end{aligned}
\tag{9}
$$

It can be seen that if we want to observe the decoupled state, $e_k$ needs to be made a function of $Me_k$, i.e.,

$$
\dot{e}_k = Me_k.
\tag{10}
$$

The following equation can be derived:

$$
0 = (HC - I)E
\tag{11a}
$$

$$
T = I - HC
\tag{11b}
$$

$$
F = A - HCA - K_1C
\tag{11c}
$$

$$
K_2 = FH
\tag{11d}
$$

If all eigenvalues of $M$ are negative, then $e(k)$ will gradually approach zero. Based on Equation (11), the value of the corresponding matrix is obtained. At this point, Equation (10) is not a function of $E$ or $d_k$, having achieved the expected decoupling of the state estimate from the unknown perturbation input.

The disturbance estimate can also be derived as:

$$
d(k|k) = (CE)^{\dagger}[y - CA\hat{x}(k|k) - CBu]
\tag{12}
$$

where $d(k|k)$ is the estimated value of the disturbance.

### 2.3. Model Predictive Controller Design

MPC control is used in this paper to implement the secondary frequency regulation control of the microgrid, and its control structure is shown in Figure 6.

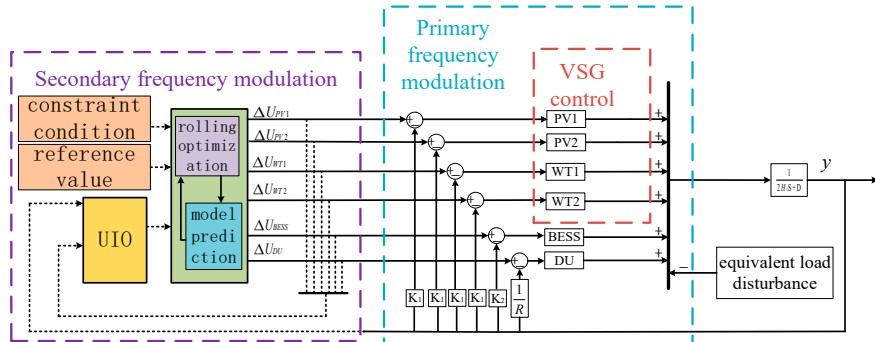

**Figure 6.** Schematic diagram of the MPC control principle.

In Figure 6, the MPC controller frequency deviation $\Delta f$ reference is 0, and the output is the power reference $\Delta U$ for each DG. The current state $x(\mathrm{k} \mid \mathrm{k})$ is observed by the proposed unknown input observer. The system state is predicted in the finite time domain based on the prediction model. The rolling optimal control is adopted to solve the optimal control variable under the satisfied constraint. The output $\Delta U$ adjusts the DG's output power to improve the system frequency response.

#### 2.3.1. Prediction Model

Considering the accuracy of the MPC and the complexity of the calculation, this paper sets the prediction time domain as $p = 12$ and the control time domain as $m = 4$. Combined with the system discrete model, the prediction values in the prediction time domain p can be obtained using the following equations:

$$Y_p(k+1 \mid k) = S\Delta x(k) + Iy(k) + S_B\Delta U(k) \tag{13}$$

$$S = \begin{bmatrix} CA & \sum_{i=1}^{2} CA^i & \sum_{i=1}^{3} CA^i & \cdots & \sum_{i=1}^{p} CA^i \end{bmatrix}^T \tag{14}$$

$$I = \begin{bmatrix} 1 & 1 & \cdots & 1 & 1 \end{bmatrix}^T_{1\times p} \tag{15}$$

$$S_B = \begin{bmatrix} CB & 0 & 0 \\ \sum_{i=1}^{2} CA^{i-1}B & CB & 0 \\ \sum_{i=1}^{3} CA^{i-1}B & \sum_{i=1}^{2} CA^{i-1}B & CB \\ \sum_{i=1}^{4} CA^{i-1}B & \sum_{i=1}^{3} CA^{i-1}B & \sum_{i=1}^{k-m+1} CA^{i-1}B \\ \vdots & \vdots & \vdots \\ \sum_{i=1}^{p} CA^{i-1}B & \sum_{i=1}^{p-1} CA^{i-1}B & \sum_{i=1}^{p-m+1} CA^{i-1}B \end{bmatrix} \tag{16}$$

where $Y_p(k+1 \mid k)$, $k = 1, 2, 3 \ldots p$.

### 2.3.2. Constraints

For generators and energy storage batteries, the available power mainly depends on their rated capacity, so the constraints of diesel engines and energy storage batteries during frequency regulation can be expressed as

$$\begin{cases} P_{DU,\min} - \overline{P}_{DU} \le \Delta U_{DU}(k) \le P_{DU,\max} - \overline{P}_{DU} \\ P_{BESS,\min} - \overline{P}_{BESS} \le \Delta U_{BESS}(k) \le P_{BESS,\max} - \overline{P}_{BESS} \end{cases} \tag{17}$$

where $P_{DU,\min}$, $P_{DU,\max}$, $P_{BESS,\min}$ and $P_{BESS,\max}$ are the upper and lower power limits of the diesel engine and energy storage unit, respectively. $\overline{P}_{DU}$, $\overline{P}_{BESS}$ is the upper dispatch power value at the current moment.

The available power for the WTs and PVs is dependent on the maximum power and deloading level $d\%$. In this paper, a 10% deloading level is chosen for each WT and PV. The estimation method for the maximum available power is provided in [36].

The available frequency regulation power constraints of the WT and PV generating sets are set as

$$\begin{cases} -d\%P_{\mathrm{MAP,WT}} \le \Delta U_{\mathrm{WT}}(k) \le d\%P_{\mathrm{MAP,WT}} \\ -d\%P_{\mathrm{MAP,PV}} \le \Delta U_{\mathrm{PV}}(k) \le d\%P_{\mathrm{MAP,PV}} \end{cases} \tag{18}$$

where $P_{\mathrm{MAP,PV}}$ and $P_{\mathrm{MAP,WT}}$ are the maximum available power of the PV and WT, respectively. $d\%$ is the deloading level. $\Delta U_{\mathrm{PV}}(k)$ and $\Delta U_{\mathrm{WT}}(k)$ are the available power of the PV and WT, respectively.

### 2.3.3. Optimization Target

The optimization objectives of the MPC control are set as follows:

$$J(x(k), \Delta U(k)) = \left\| \Gamma_y x(k+1 \mid k) \right\|^2 + \left\| \Gamma_u \Delta U(k) \right\|^2 \tag{19}$$

The objective function includes state variables and control variables, which can be optimally solved by considering these two sets of variables, where $\Gamma_y$ and $\Gamma_u$ are the weight matrices of the state variable and output variable, respectively.

$$\begin{cases} \Gamma_y = diag\{ \; \alpha \quad \alpha \quad \alpha \quad \cdots \quad \alpha \; \} \\ \Gamma_u = diag\{ \; \beta PV1 \quad \beta PV2 \quad \beta WT1 \quad \beta WT2 \quad \beta DU \quad \beta BESS \; \} \end{cases} \tag{20}$$

where $\Gamma_y$ is used to penalize the system frequency deviation and $\Gamma_u$ is used to penalize the control output, i.e., the output power of the generation unit; $\alpha$ and $\beta$ are the corresponding penalty factors.

To make full use of renewable energy, it is better to prioritize the output power of the RESs participating in frequency regulation. We set the penalty factors of the PVs and WTs to be lower than those of the battery and diesel units. In this paper, $\alpha = 1.395$; the wind and photovoltaics had a penalty factor of $\beta_{WT} = \beta_{PV} = 0.2432$, and the diesel unit and battery had a penalty factor of $\beta_{DU} = \beta_{Bess} = 0.3236$.

The diesel and storage units have a larger penalty factor than the WTs and PVs. This treatment can prioritize the wind and photovoltaic units for participation in secondary frequency regulation.

### 2.3.4. Constraint-Containing Optimization Problem Solving

Due to the existence of constraints, the optimal solution of the objective function cannot be directly obtained. Therefore, it is necessary to convert the MPC optimization problem with variable constraints into a quadratic programming problem. The prediction equation is brought into the objective function and, for the optimization problem, the objective function is simplified as:

$$J = \Delta U(k)^{\mathrm{T}} (S^T \Gamma_y^T \Gamma_y S^T + \Gamma_u^T \Gamma_u) \Delta U(k) - (2S^T \Gamma_y^T \Gamma_y E_p(k+1 \mid k))^{\mathrm{T}} \Delta U(k) \tag{21}$$

Converting the constraints into inequality form, we obtain:

$$C\Delta U(k) \geq B(k+1|k) \tag{22}$$

To solve the optimization problem in quadratic programming form, we define new variables, i.e., where $R$ is the reference value matrix.

$$\rho = \begin{bmatrix} \Gamma_y\big(Y_p(k+1) - R(k+1)\big) \\ \Gamma_u\Delta U(k) \end{bmatrix} \tag{23}$$

The optimization problem then becomes min $\rho^T\rho$, and the extreme value condition of $\rho^T\rho = (Az - b)^T(Az - b)$ is derived from the derivative to obtain the extreme value solution.

$$Z^* = (A^TA)^{-1}A^Tb \tag{24}$$

The optimal control sequence that is obtained by solving is as follows:

$$\Delta U^*(k) = K_{mpc}(0 - Y_p(k+1 \mid k)) \tag{25}$$

$$K_{mpc} = \left(\mathcal{S}_B^T\Gamma_y^T\Gamma_y\mathcal{S}_B + \Gamma_u^T\Gamma_u\right)^{-1}\mathcal{S}_B^T\Gamma_y^T\Gamma_y \tag{26}$$

Only the first element of the optimal control sequence is taken as the control output.

Based on the above discussion, the flow chart of the MPC controller designed in this paper is shown in Figure 7.

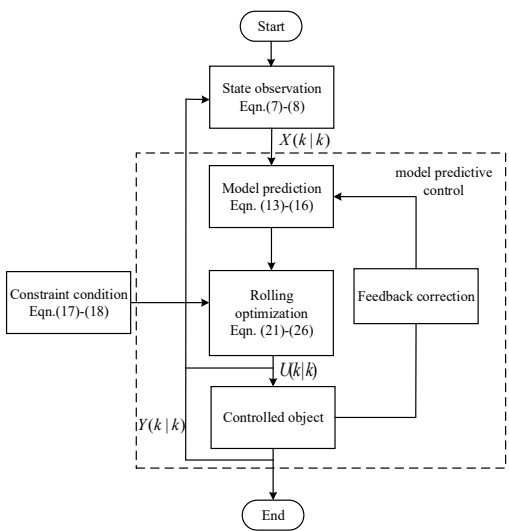

**Figure 7.** MPC control flow chart.

The state observer obtains the current system state $X(k|k)$ according to the input system output $Y(k|k)$ and control command $U(k|k)$ and passes it to the MPC controller. The MPC controller predicts the system state in the prediction time domain through the prediction model according to the current state. The optimization is solved according to the optimization objective and set real-time variable constraints. The first control command in the optimal control sequence is sent to the control object to complete the optimal control.

### 2.4. Equivalent Load and Wind Power, Photovoltaic, Load Uncertainty Modeling

To facilitate analysis, the WT and PV output power can be equivalent to a 'negative' load. The equivalent load power of the system is the difference in power between the load and PV as well that of the WT power. The equivalent load power of the system is

$$\Delta P_E = \Delta P_L - \Delta P_W - \Delta P_S \tag{27}$$

where $\Delta P_E$ is the equivalent load power of the system, $\Delta P_L$ is the load power, $\Delta P_W$ is the WT power, and $\Delta P_S$ is the PV power. The equivalent load perturbation is represented as an unknown input perturbation by the *B* array. The specific values are shown in Appendix A.

The wind speed, irradiance, and system load fluctuate. In this study, the wind speed, irradiance, and load are obtained according to statistical rules.

Irradiance is usually approximated to obey the lognormal distribution. The probability distribution function is as follows:

$$f(I_t|\mu_t, \sigma_t) = \frac{1}{I_t \mu_t \sqrt{2\pi}} \exp\left(\frac{-(\ln I_t - \sigma_t)^2}{2\sigma_t^2}\right) \tag{28}$$

where $I_t$ is the solar irradiance at time $t$ and $\mu_t, \sigma_t$ is the parameter of the lognormal distribution function.

Wind speed is described as the three-parameter Burr distribution function, which is expressed as

$$f(v_t|a_t, c_t, k_t) = \frac{\left(\frac{k_t c_t}{a_t}\left(\frac{v_t}{a_t}\right)^{c_t-1}\right)}{\left(1 + \left(\frac{v_t}{a_t}\right)^{c_t}\right)^{k_t+1}} \tag{29}$$

where $v_t$ is the wind speed at time $t$;$a_t, c_t, k_t$ are the scale parameter, first shape parameter, and second shape parameter of the Burr distribution at time $t$, respectively. According to the generated wind speed and irradiance data, the output power of the WT and PV is obtained.

The load fluctuation can be expressed by the normal distribution function that is shown in the following equation:

$$f(P) = \frac{1}{\sqrt{2\pi}\sigma_p} \exp\left(-\frac{(P - \mu_p)^2}{2\sigma_p^2}\right) \tag{30}$$

where $\sigma_p$ is the standard deviation of active power; $\mu_p$ is the mean value of active power.

Details of the probabilistic model and the remaining parameters can be found in [37].

## 3. Results

To verify the effect of the proposed control strategy, according to Figure 1, this paper built a simulation model of a microgrid consisting of wind turbines and photovoltaic, diesel, and energy storage batteries. The traditional PI control, conventional MPC control, and proposed MPC control are the three control methods used for secondary frequency regulation. In the study, all parameters are normalized.

### 3.1. Wind Speed, Solar Irradiance, and Load Fluctuation Scenarios

According to the probability model proposed in Section 2.4, take $\mu_t = 0.01$ $\sigma_t = 0.02$ $a_t = 2.15, c_t = 2, k_t = 0.5$ $\sigma_p = 0.08$ $\mu_p = 0$. The generated RESs/DGs power data and equivalent load are shown in Figure 8.

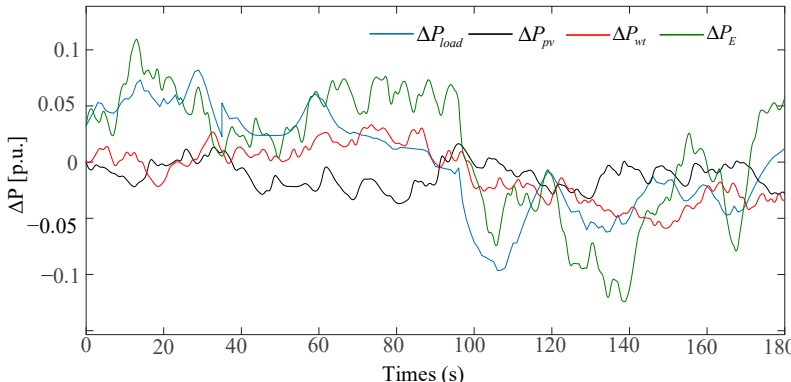

**Figure 8.** Random disturbance of WT, PV, loads, and equivalent load.

Figure 9 shows the observation results of the decoupled observer and observation error. Figure 9a shows the observed results of the equivalent load perturbations. In the equivalent load fluctuation smooth range, which ranges from 160 s to 180 s, the observation performance is accurate. In the high fluctuation range, which ranges from 40 s to 80 s, the observation performance is degraded. This means that the observation error is 1%. Overall, the observed value for the proposed observer is close to the real value, although the load perturbations quickly vary.

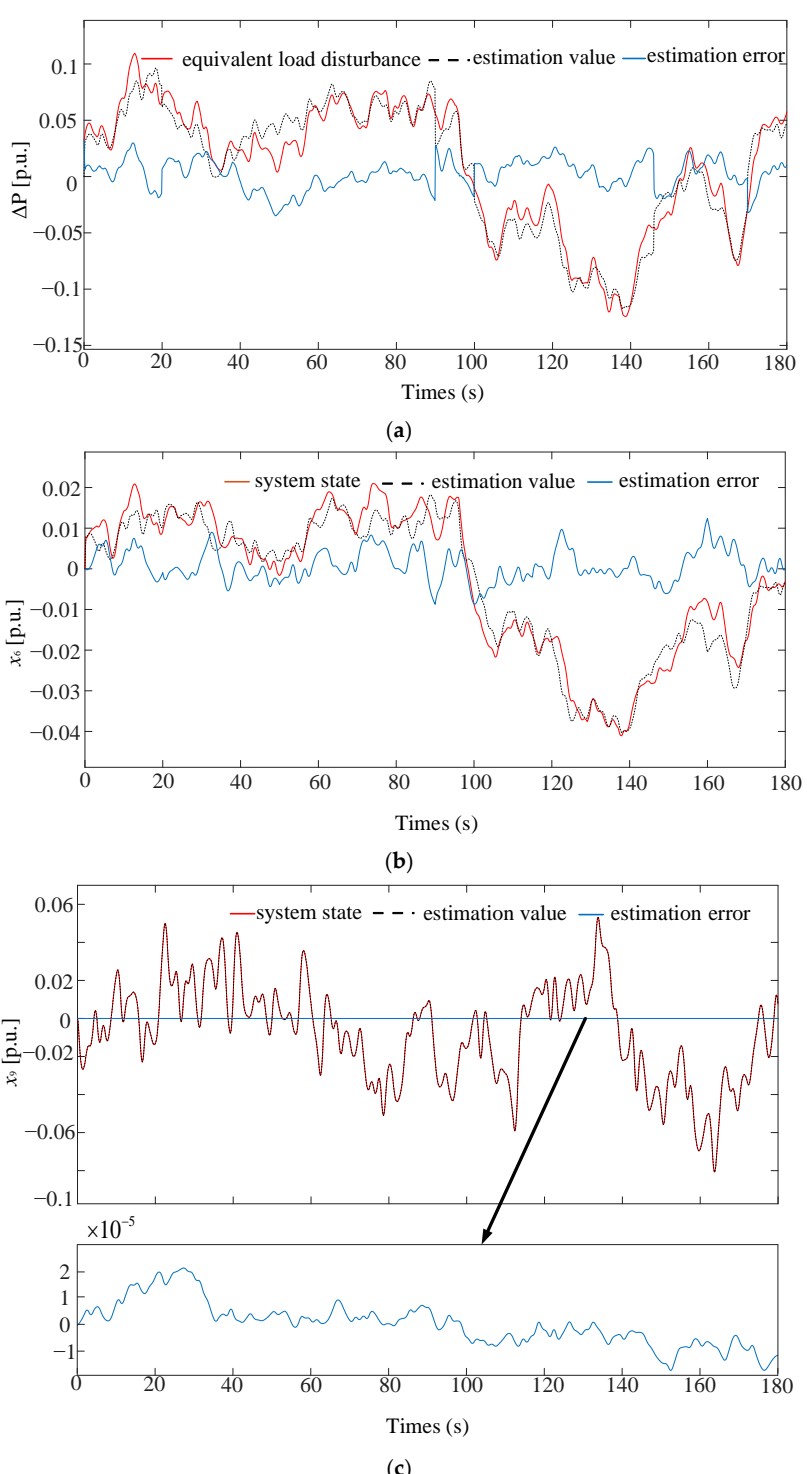

**Figure 9.** *Cont.*

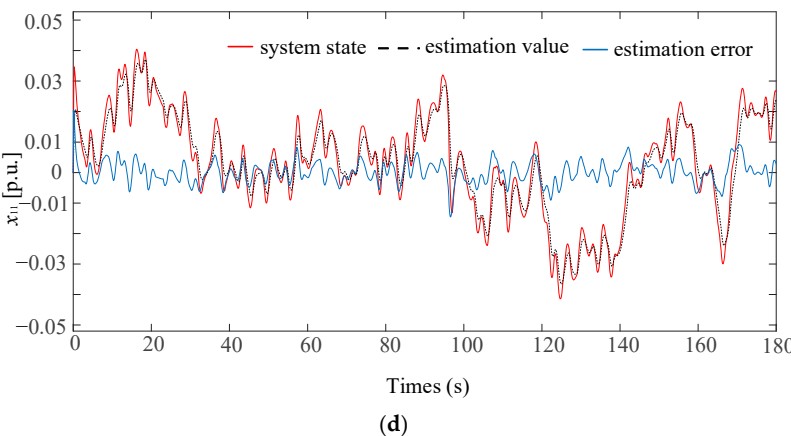

(**d**)

**Figure 9.** (**a**) Equivalent load estimation value; (**b**) system state $x_6$ estimation value; (**c**) system state $x_9$ estimation value; (**d**) system state $x_{11}$ estimation value.

Figure 9b shows the observation results of $x_6$, that is, the output power of the WT. Figure 9c shows the observation results of $x_9$, that is, the output power of the energy storage battery, and Figure 9d shows the observation results of $x_{11}$, that is, the diesel engine angular velocity. $x_6$ is the output of the second-order system, and $x_9$ is the output of the first-order system; it can be seen that, compared with the second-order system, the observer has a better observation effect on the first-order system.

Figure 10 is the power instruction value of the controller.

The upper and lower limits of power of the diesel engine and energy storage battery are fixed; however, due to the fluctuations in the irradiance and wind speed, the upper and lower limits of PV and WT fluctuate.

Observing Figure 10, we see that the output of the PI controller is proportionally distributed to the DGs. The output power of each DG is approximate.

While there are two MPC controls, by solving the quadratic programming problem, the optimal power of each DG is obtained. Due to the setting of the weight coefficient, the outputs of PV and WT will be higher than that of the diesel engine and energy storage battery. Across some time periods, such as from 10 s to 30 s, the PV and WT outputs reach their maximum values but do not cross the limit due to the real-time variable constraint setting.

Figure 11 is the frequency response result of Case 1.

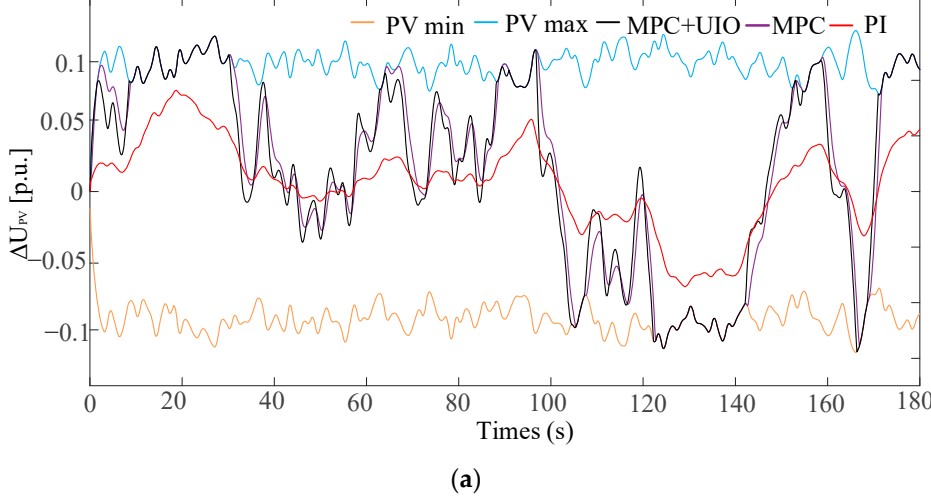

(**a**)

**Figure 10.** *Cont*.

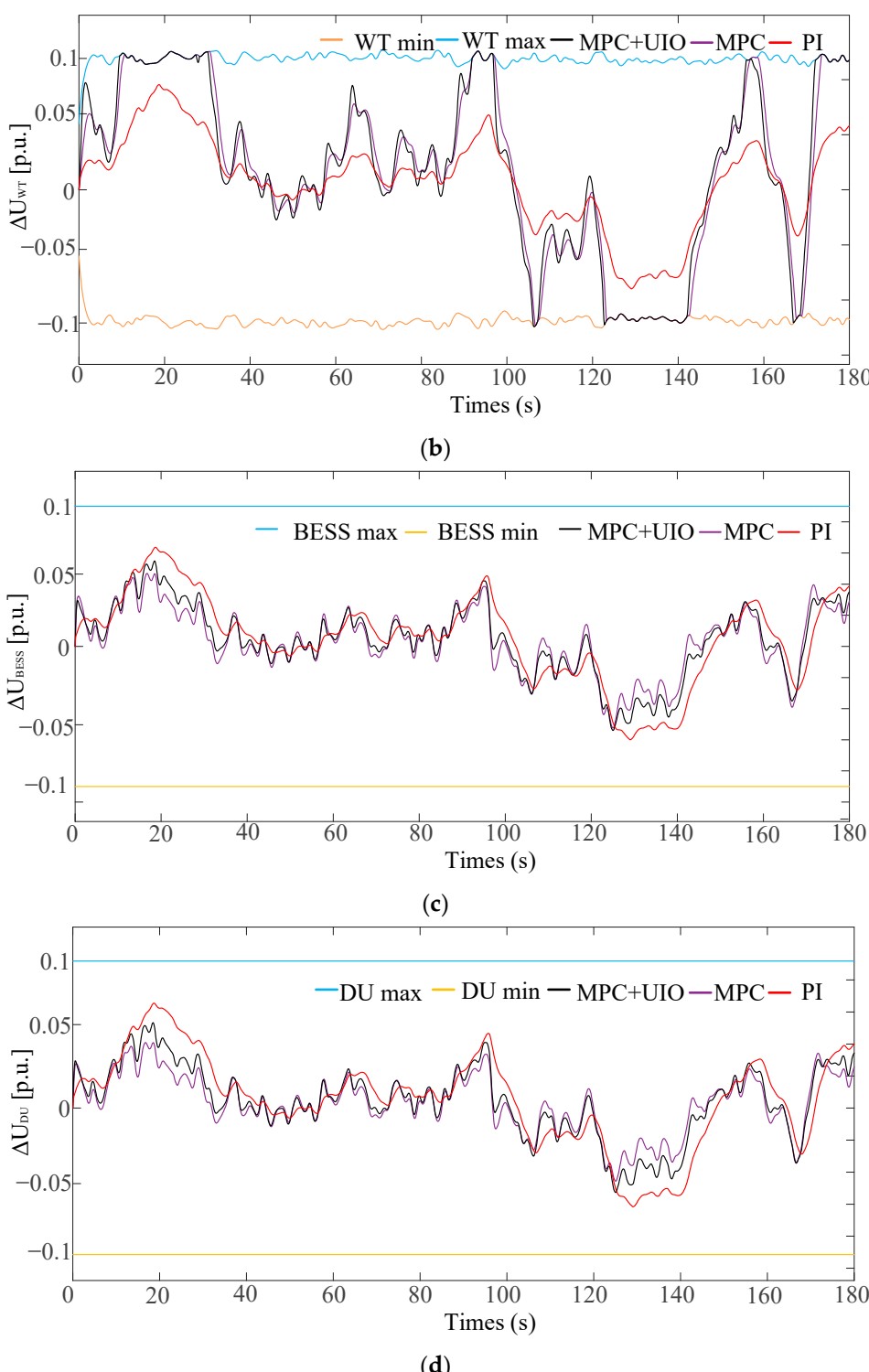

**Figure 10.** Controller instruction value in Case 1. (**a**) PV power output, (**b**) WT power output, (**c**) energy storage output, (**d**) diesel engine output.

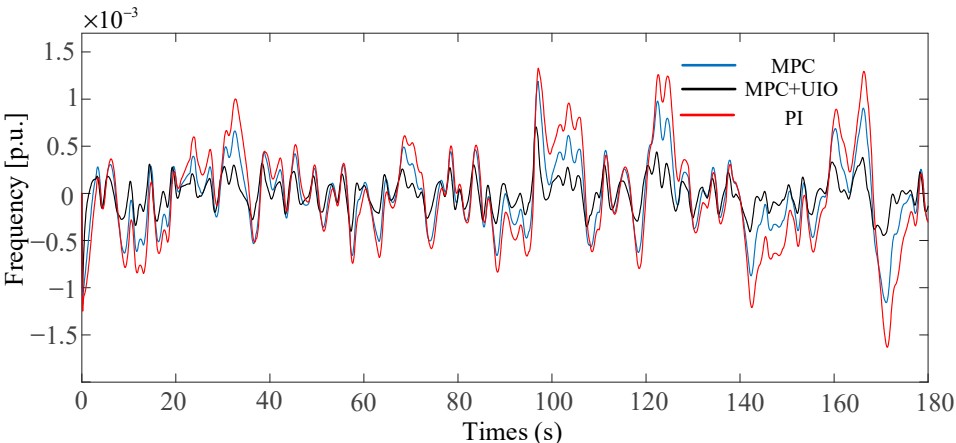

**Figure 11.** System frequency in Case 1.

As seen in Figure 11, among the three control methods, two MPC methods had a better frequency control effect than the PI control because MPC uses real-time rolling optimization and can optimally distribute the restorative power between all DGs. The proposed control method has the best frequency response because the used unknown input observer has an accurate observed value that is beneficial for model prediction in the prediction time domain. The standard deviation of the PI control is $7.174 \times 10^{-3}$, MPC is $5.232 \times 10^{-3}$, and UIO + MPC is $3.651 \times 10^{-3}$.

### 3.2. Severe Fluctuation Scenarios of Wind Speed, Solar Irradiance, and Load

In some severe weather conditions, wind speed and solar irradiance may rapidly change, such as through the rapid shading of clouds, sudden changes in wind speed, etc. Take $\mu_t = 0.01 \; \sigma_t = 0.05 \; a_t = 1.22, c_t = 1.1, k_t = 2 \; \sigma_p = 0.15 \; \mu_p = 0$. This generates a set of fluctuations and more violent random perturbations, and the data are shown in Figure 12 below.

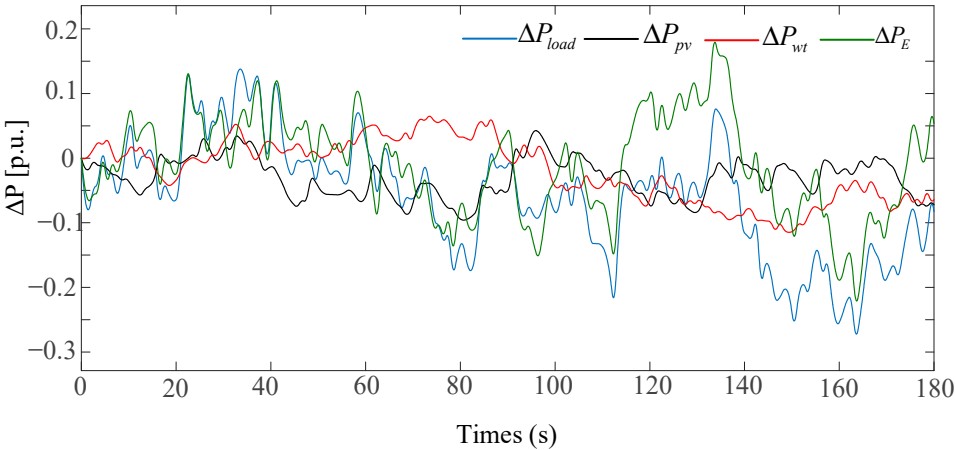

**Figure 12.** Random disturbance of the WTs, PVs, loads, and equivalent load.

As shown in Figure 13, although the fluctuation is intensified, the observer can still obtain an accurate estimate. The average observation error is 1.5%. The observation effect of the first-order system is better than that of the second-order system.

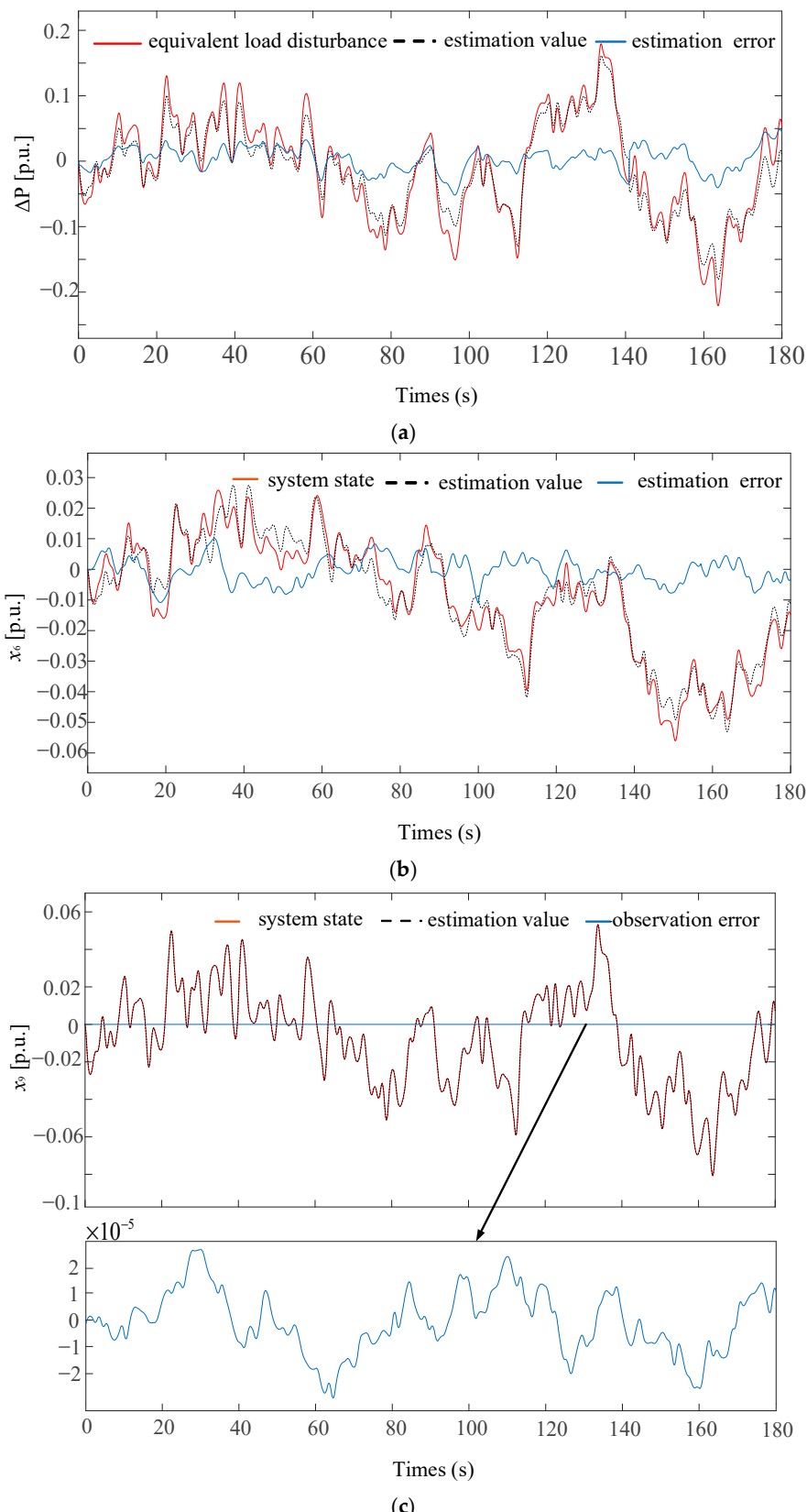

**Figure 13.** *Cont.*

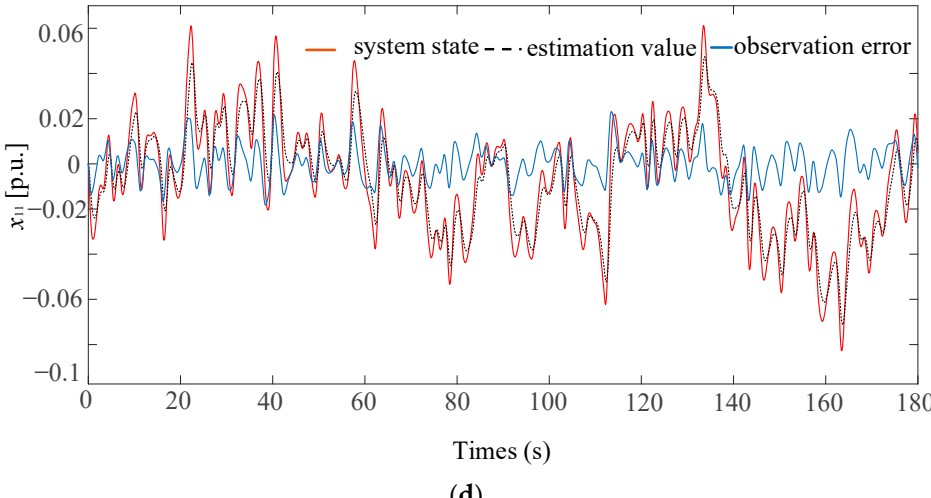

**(d)**

**Figure 13.** Observer output in Case 2. (**a**) Equivalent load estimation value, (**b**) system state $x_6$ estimation value, (**c**) system state $x_9$ estimation value, (**d**) system state $x_{11}$ estimation value.

Similarly, the traditional PI control allocates the recovery power proportionally to each DG. Subfigures a–d show that the PV and WT reach saturation at many time points. However, the MPC control is the optimal control under the constraint; the MPC will increase the output of other DGs when some RESs reach the power limit. Therefore, in this time frame, the output powers of the diesel and storage battery will be increased appropriately to ensure frequency regulation performance.

Figure 15 is the frequency response result of Case 2. Among the three control methods, the proposed method has the best frequency control effect. The standard deviation of the PI control is $9.817 \times 10^{-3}$, MPC is $7.613 \times 10^{-3}$, and the UIO + MPC is $4.612 \times 10^{-3}$.

Figure 14 is the output of the controller in Case 2.

To reflect the performance of the proposed controller more intuitively in the secondary frequency regulation control response, Table 3 depicts the maximum frequency deviation and standard deviation of the three controllers.

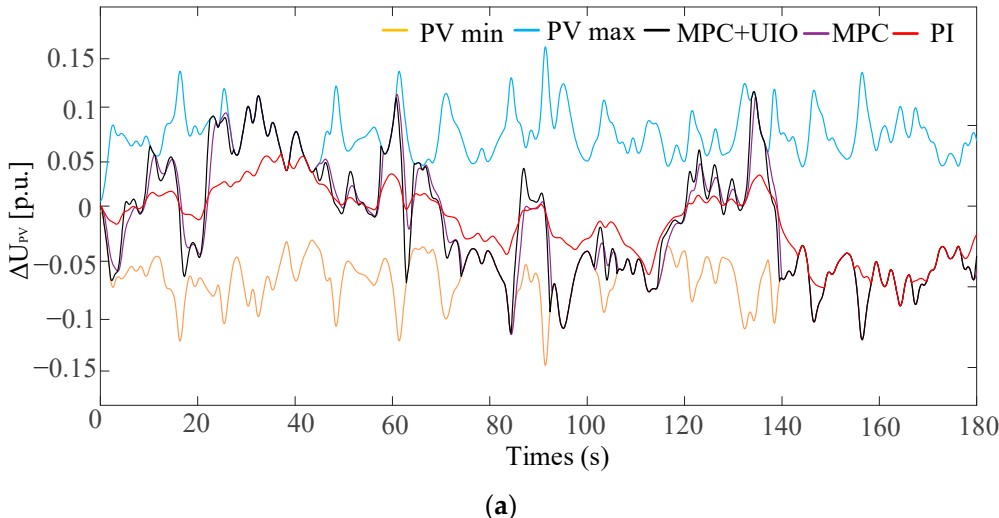

**(a)**

**Figure 14.** *Cont.*

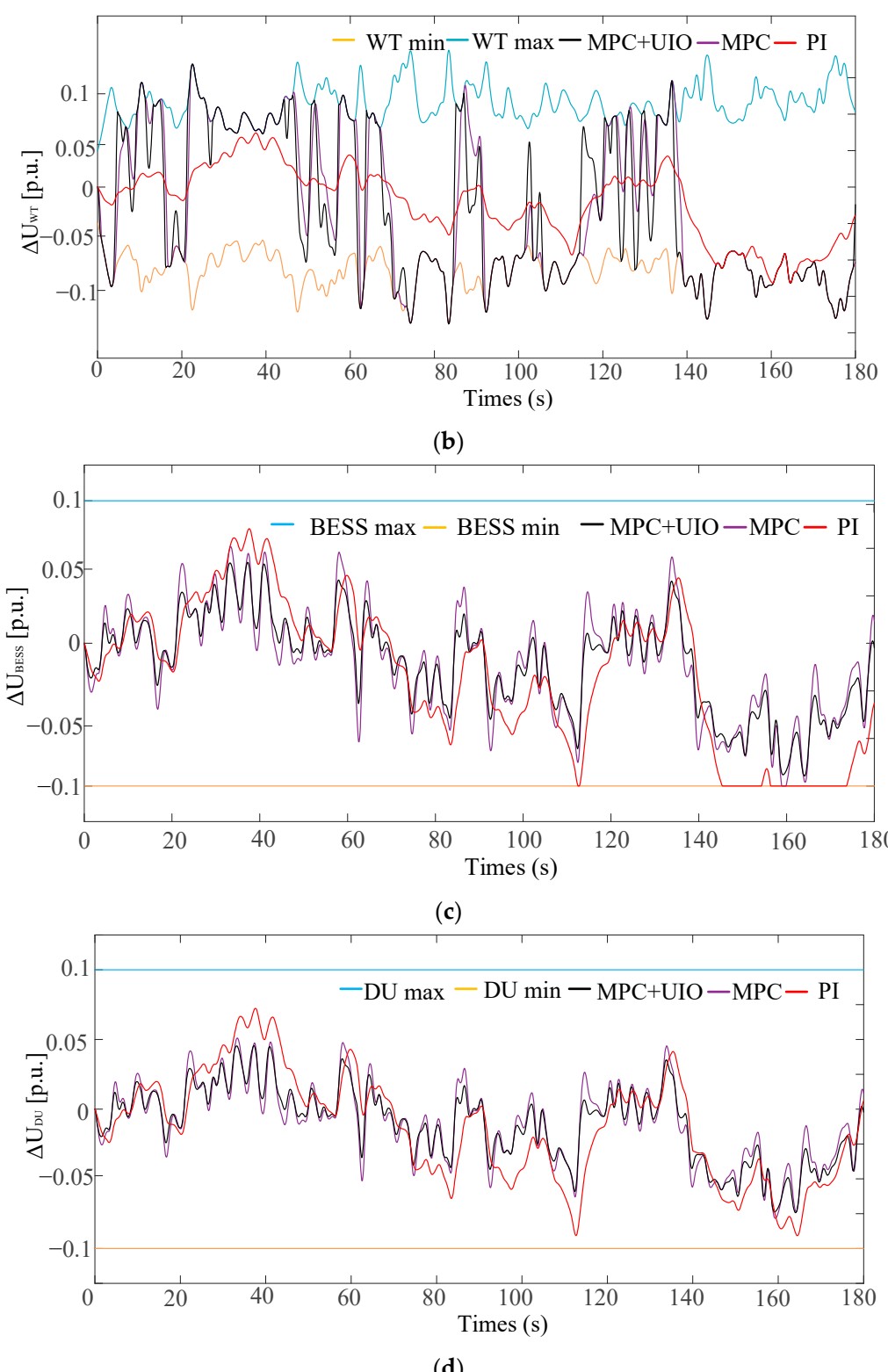

**Figure 14.** Controller instruction value in Case 2. (**a**) PV power output, (**b**) WT power output, (**c**) energy storage output, (**d**) diesel engine output.

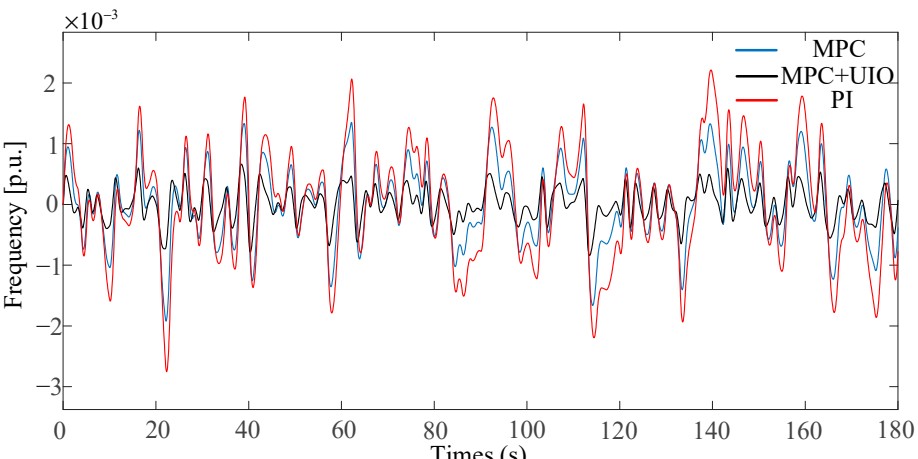

**Figure 15.** System frequency in Case 2.

**Table 3.** Frequency control effects of the three controllers.

|  | Control Method | Standard Deviation | Maximum Error | Unit |
|---|---|---|---|---|
| | PI | $7.174 \times 10^{-3}$ | $1.56 \times 10^{-3}$ | p.u. |
| Case 1 | MPC | $5.232 \times 10^{-3}$ | $1.22 \times 10^{-3}$ | p.u. |
| | MPC + UIO | $3.651 \times 10^{-3}$ | $0.71 \times 10^{-3}$ | p.u. |
| | PI | $9.817 \times 10^{-3}$ | $2.73 \times 10^{-3}$ | p.u. |
| Case 2 | MPC | $7.613 \times 10^{-3}$ | $1.94 \times 10^{-3}$ | p.u. |
| | MPC + UIO | $4.612 \times 10^{-3}$ | $0.84 \times 10^{-3}$ | p.u. |

## 4. Conclusions

This paper proposes an MPC microgrid secondary frequency control method by incorporating an unknown input observer. RESs adopt a deloading VSG control and participate in centralized secondary frequency regulation. The proposed unknown input observer, when decoupled, observes the unknown equivalent load input and system state, respectively.

Compared with the conventional PI control, the results show that the proposed controller reduces the standard deviation by 49% in the case of small perturbations and by 53% in the case of large perturbations. Benefiting from the improvement in the observation effect, when compared with the normal MPC control, the results show that the proposed controller reduces the standard deviation by 31% in the case of small perturbations and by 38% in the case of large perturbations.

The proposed method improves the insufficient inertia of the RES high-permeability microgrid and enables the RESs to participate in secondary frequency regulation. The unknown input observers improve the observation accuracy, thus improving the control effect of MPC. The MPC controller gives priority to the RES output, which reduces the consumption of the energy storage battery and diesel engine. The proposed control strategy in this paper has a better control effect on microgrid frequency control.

In this paper, the influence of randomness on the MPC algorithm is not considered; this may affect the control effect of the controller. In subsequent research, the stochastic model predictive control method will be studied to further enhance the frequency control effect under the high stochastic power perturbation scenario, for example, in chance-constrained MPC and tree-based MPC applications.

**Author Contributions:** Conceptualization, Z.Z.; methodology, Z.Z., X.Z. and C.Z.; software, X.Z. and C.Z.; validation, Z.Z.; investigation, Z.Z. and X.Z.; resources, C.Z.; writing—original draft preparation, Z.Z.; writing—review and editing, X.Z. and C.Z.; visualization, C.Z.; supervision, C.Z.; project administration, Z.Z. All authors have read and agreed to the published version of the manuscript.

**Funding:** This research was funded by Jilin Province Science and Technology Development Plan Project under grant ID 20190201289JC.

**Data Availability Statement:** The data are available in a publicly accessible repository.

**Conflicts of Interest:** The authors declare no conflict of interest.

**Appendix A**

$$
A = \begin{pmatrix}
-\frac{D_1}{J} & -\frac{K_1}{J} & 0 & 0 & 0 & 0 & 0 & 0 & 0 & 0 & 0 & -\frac{K_1 K}{J} \\
K_2 & 0 & 0 & 0 & 0 & 0 & 0 & 0 & 0 & 0 & 0 & -2\pi K_2 \\
0 & 0 & -\frac{D_1}{J} & -\frac{K_1 K_2}{J} & 0 & 0 & 0 & 0 & 0 & 0 & 0 & -\frac{K_1 K}{J} \\
0 & 0 & K_2 & 0 & 0 & 0 & 0 & 0 & 0 & 0 & 0 & -2\pi K_2 \\
0 & 0 & 0 & 0 & -\frac{D_1}{J} & -\frac{K_1 K_2}{J} & 0 & 0 & 0 & 0 & 0 & -\frac{K_1 K}{J} \\
0 & 0 & 0 & 0 & K_2 & 0 & 0 & 0 & 0 & 0 & 0 & -2\pi K_2 \\
0 & 0 & 0 & 0 & 0 & 0 & -\frac{D_1}{J} & -\frac{K_1 K_2}{J} & 0 & 0 & 0 & -\frac{K_1 K}{J} \\
0 & 0 & 0 & 0 & 0 & 0 & K_2 & 0 & 0 & 0 & 0 & -2\pi K_2 \\
0 & 0 & 0 & 0 & 0 & 0 & 0 & 0 & -\frac{1}{T_{bess}} & 0 & 0 & 0 \\
0 & 0 & 0 & 0 & 0 & 0 & 0 & 0 & 0 & -\frac{1}{T_{du1}} & 0 & -\frac{1/R}{T_{du1}} \\
0 & 0 & 0 & 0 & 0 & 0 & 0 & 0 & 0 & \frac{1}{T_{du2}} & -\frac{1}{T_{du2}} & 0 \\
0 & \frac{1}{6H} & 0 & \frac{1}{6H} & 0 & \frac{1}{6H} & 0 & \frac{1}{6H} & \frac{1}{6H} & 0 & \frac{1}{6H} & -\frac{D}{H}
\end{pmatrix}
$$

$$
B = \begin{pmatrix}
\frac{k_1}{J} & 0 & 0 & 0 & 0 & 0 & 0 \\
0 & 0 & 0 & 0 & 0 & 0 & 0 \\
0 & \frac{k_1}{J} & 0 & 0 & 0 & 0 & 0 \\
0 & 0 & 0 & 0 & 0 & 0 & 0 \\
0 & 0 & \frac{k_1}{J} & 0 & 0 & 0 & 0 \\
0 & 0 & 0 & 0 & 0 & 0 & 0 \\
0 & 0 & 0 & \frac{k_1}{J} & 0 & 0 & 0 \\
0 & 0 & 0 & 0 & 0 & 0 & 0 \\
0 & 0 & 0 & 0 & \frac{1}{T_{bess}} & 0 & 0 \\
0 & 0 & 0 & 0 & 0 & \frac{1}{T_{du1}} & 0 \\
0 & 0 & 0 & 0 & 0 & 0 & 0 \\
0 & 0 & 0 & 0 & 0 & 0 & \frac{1}{H}
\end{pmatrix}
$$

$$
C = \begin{pmatrix} 0 & 0 & 0 & 0 & 0 & 0 & 0 & 0 & 0 & 0 & 0 & 1 \end{pmatrix}
$$

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
