# Peer review of "Model Predictive Secondary Frequency Control for Islanded Microgrid under Wind and Solar Stochastics"

_electronics, doi:10.3390/electronics12183972_

Round 1

Reviewer 1 Report

The paper presents a secondary frequency control strategy using model predictive control for islanded microgrids. The authors highlight importance of frequency control in the context of stochastic power disturbance.

The paper covers the appropriate topic and is sustained by actual and relevant references, without an excessive number of self-citations. The introduction section may be improved in the first paragraph, where is a summary of 6 cited papers. In-depth conclusions from the analyzed papers should be included so that this paragraph have sense (Microgrids can be operated islanded, where the primary objective is to maintain the balance between sources and loads.RESs/DGs used inverter-interfaced, which lack nature inertia and add more power fluctuations and disturbances[2-5]. It brings more challenges for the frequency regulation of the microgrid[6].)

The experimental design of the MPC is appropriate to test the hypothesis of the paper.

The figures/tables/images/schemes are appropriate, easy to interpret and understand and relevant for the study performed.

The Figure 7 presents the MPC control flowchart. It would be useful to include a short description of the process based on the flowchart for constraint-containing optimization problem solving.

The conclusions are consistent with the evidence and arguments presented in the paper.

The authors should also highlight how the paper brings a new approach for the subject (predictive control for microgrids) compared to other papers published in the field (for example https://doi.org/10.1016/j.asej.2020.12.007 or https://doi.org/10.3390/en16134851)

The English language is appropriate.

Author Response

尊敬的审稿人,感谢您给我们机会提交稿件的修订稿,我们仔细研究了您的意见,并根据您的建议逐项仔细修改了论文。请参阅附件

Reviewer 2 Report

Decision: Major Revision

Summary:

I understood that the importance of controlling microgrids. Therefore, I think that there would be much interest in development a targeted control model.

However, please describe more clearly why you are targeting Islanded microgrids and how this research differ from other studies. Also, please clarify what size of microgrid the model is designed for, including the need for it.

The readability of the document is poor, with typos and chapter titles at the end of the page. Please revise the information significantly.

Each part

Line number

Please correct the line numbering.

Result

10p Line.2-11p Line.10 is methodology and should be moved to METHODOLOGY.

5.1. Wind speed, solar irradiance, and load fluctuation scenarios etc.

How did you decided parameters like =0.01etc. What conditions are these parameters intended for? Please add a explanation.

Fig.10 (a) etc.

Please change the figure to show the difference between items that are not predicted (like MIN/MAX) and items that are predicted.

Conclusion

The current document is a technical report. Please describe your discussion of the results of this study, referencing the results and limitations of this study.

Author Response

Dear reviewer,Thank you for giving us the opportunity to submit a revised draft of the manuscript,We have carefully studied your comments and carefully revised the paper item by item according to your suggestions.Please see the attachment

Reviewer 3 Report

This paper presents a new secondary control predictive model that integrates an unknown input observer into an isolated microgrid. In this microgrid configuration, Renewable Energy Sources (RESs) and Distributed Generators (DGs) are employed to help control the virtual synchronous generator. The use of an unknown input observer serves to accurately estimate the states of the system, as well as the stochastic perturbations of energy from the RESs/DGs and the load. The distributed restorative capacity of each DG is determined by solving an optimal quadratic programming problem, incorporating variable constraints.

The simulation results provide convincing evidence that the proposed approach significantly increases the speed of system frequency restoration, thus leading to decreased frequency deviations when juxtaposed with conventional secondary control methodologies.

The manuscript exhibits a commendable organizational structure. Both its introductory section and subsequent segments on methodology and results reach a satisfactory standard. The applicability demonstrated in the results adds a layer of interest. Consequently, I am inclined to recommend its publication, subject to minor revisions to correct typographical and linguistic errors that have been identified in the text.

Minor editing of English language required

Author Response

(The authors gave the same response as above.)

Reviewer 4 Report

1. Don't cited the reference number more than once, all cited reference number should be in order. 

2. Please include results of bridge converter and inverter results of input and output of voltage and current. 

3. Please mentioned inverter PWM waveforms and explain. 

4. Please use filter results after inverter output. 

5. Please use inverter results with and without load. 

6. Minor English corrections are required. 

Minor English corrections are required. 

Author Response

(The authors gave the same response as above.)

Reviewer 5 Report

 I have carefully studied Model Predictive Secondary Frequency Control for islanded  microgrid under wind and solar stochastics And have the following observations for the author(s):

-The abstract is too short and does not provide useful details to the readers. The authors have done a great job but they have not explained their work in the Abstract. Please follow some previous papers and provide background, findings, methodology, and brief policies in this section to make it more attractive to readers

-please, provide those theoretical reasoning that underpin the direction of the authors research investigations and the econometric model used in the study

- The limitation of the study is missed.

-the novel contribution of this work is missed

Author Response

(The authors gave the same response as above.)

Reviewer 6 Report

The paper introduces an important topic to integrate different sources in the grid. The work is promising after some major considerations as outlined below:

1- English languge needs extensive revision. For example, the 2nd statement in the abstract is not written properly. Also title 2.1. Other comments are available in the attached annotated PDF.

Test editing requires some effort. For example, in many places along the manuscript, spaces should be added as underlined in the first page of annotated PDF. Please apply for the whole manuscript.

Symbols are written in different fonts/formats and also in different font sizes. Please unify.

After each equation, either we have dot to end the statement or comma to continue with the next line that should start with a small letter. This should be considered in the whole manuscript.

Some symbols are not declared and others have declarations without appear in the text.

Formula/Equation/Eq() are used in the manuscript. Please unify.

The paper title mentioned the wind and solar stochastics. But nothing in the paper are analyzing this stochastic nature.

Please see the attached annotated PDF file.

Regards

Major revisions are required.

Author Response

(The authors gave the same response as above.)

Round 2

Reviewer 2 Report

Decision: Major Revision

Summary:

I think that the predictive control of island microgrid is a very important theme. However, in this study, there are many difficult understand parts whether the calculations were done appropriately, such as the parameters were not specified in the explanation of the calculations.

Since the subscripts in the formulas are not defined, it is not clear what is being calculated. Especially, for X9, the calculation produced a result with a prediction error of zero, which is not a normal result in a calculation.

Chapters 2-5 explain the Methodology part, but there are many expressions that are difficult for readers to understand, such as calculation settings mixed in with the Result parts.

From the above, it is necessary to make major revisions so that the readers could understand the contents.

Each part

Equations

Please provide explanations for all characters in the formula, including subscripts. Please also provide the units of measure.

Chapter 2-5

These Chapters should be combined into one Chapter because they are methodological explanations. Also, 11p Line.308-316 is a setup for the calculations. Please describe them in Methodology, not in Results.

Figure.9

I think the units are [p.u.], am I wrong? Some of these figure use [pu].

If you use (), it looks like a variable that has an effect on X, so please modify it with [] or something similar.

I do not understand what is specifically indicated by X9 from the text. Therefore, I do not understand the prediction that Error is 0 in (c). Please correct the legend and the figure as they are different in some figures.

Table.1, Tabla2

There is a parameter "Load1, Load2" but it is not mentioned in the formula of the methodology. Please add the explanations of these parameters. In addition, there are several other parameters that are not explained in the formulas, so please modify the formulas to match these parameters.

P13 Line.319-320

You wrote "The probability model and the remaining parameters can be found in paper [38]." . We are not sure if the reference setting is appropriate for this study, so please provide a reason as to why you adopted the values from this reference.

This section should be move in Methodology.

Table.3 etc.

This study uses [p.u.] as the unit of measure. Is this in units of [p.u.] at the time of the calculation input, or is it [p.u.] by calculation within the model? Please add an explanation.

I do not have the impression that the quality of the English is poor, but there are many points that are not well explained.

Author Response

(The authors gave the same response as above.)

Reviewer 4 Report

I pleased to inform that the current version of manuscript accepted for publication.

Author Response

We sincerely thank you for your recognition of our paper. Wish you a happy day.

Reviewer 6 Report

It is now Ok from my point of view.

Author Response

(The authors gave the same response as above.)

Round 3

Reviewer 2 Report

Decision: Minor Revision

Summary:

I think that this imporving has be made the content easier to understand. In particular, I think the part of Figure 13 (c), where the results are easier to understand, is good. I think that it is easy for many readers to understand and worthy of publish.

In Lines 165-170, Tdu1 and Tdu2 are different from those in Table 2. Also, H is Hs in Fuigure.4 and Ht in Table 2. Please unify the words in the formulas.

The added sentences are spoken style, so please recheck your paper again.

There are colloquial wording in some sentences.

Author Response

(The authors gave the same response as above.)
